# Gender and Empowerment by Nursing Students: Representations, Discourses and Perspectives

**DOI:** 10.3390/ijerph20010535

**Published:** 2022-12-28

**Authors:** Isabela Nogueira, Gabriela Spagnol, Fernanda Rocha, Maria Helena Lopes, Dalvani Marques, Debora Santos

**Affiliations:** Faculty of Nursing, The State University of Campinas, São Paulo 13083-970, Brazil

**Keywords:** nursing education research, feminism, gender studies, empowerment, history of nursing, gender equity

## Abstract

Nursing history is marked by stigmas of gender, race and class. Nowadays, this scenario is evidenced by the social disqualification of the profession and biomedical and male supremacy. Nevertheless, the profession has the potential to change this paradigm with an intersectional approach. The current study aims to understand how the relationships of gender, feminism and empowerment are experienced by nursing students at a Brazilian public university. This is a qualitative study, exploratory-explanatory, with the application of interviews with nursing students in their five years of training. The chosen method of analysis was the Discourse of the Collective Subject based on the central ideas categorized after the interviews: (a) Profession—female and stigmatized due to its historical construction influenced by religiosity and moral; (b) Formation—far from gender relations by the perpetuation of stereotypes; and (c) Perspectives—empowerment of the profession if close to the feminist movement. The students’ discourse alert to the historical reflexes of oppressive ideological mechanisms of women and nursing in their ongoing professional training, claiming transversal learning spaces for the critical expansion of gender awareness and consequent empowerment of nursing in a feminist and intersectional perspective.

## 1. Introduction

The feminist debate on social inequalities imposed by gender relations has been incorporated by international and national health organizations since the 1980s, as one of the priority needs for promoting equity in health [1]. Despite this priority, the discussion on gender in the field of nursing was leveraged in 2016, with the emergence of the international movement Nursing Now, which aims to promote global support and category empowerment.

Allied to this movement, the celebrations for the 2020 International Year of Nursing and Midwifery match with the recognition and decisive action of nursing in the face of the challenges arising from the COVID-19 pandemic [2]. This category represents the largest group among health professionals, with 59% of the global health workforce, which totals 27.9 million nursing professionals [3]. Among these, nine out of ten are female [3], demonstrating the relevance of the debate on the gender perspective for a better understanding nursing practice and education.

From a historical point of view, the social role of nursing closely followed that destined for women over the centuries. For the American historian Joan Scott [4], social inequalities bring a strong mark of gender relations, defined according to cultural attributes imposed on the masculine and feminine that place women as “second-class citizens”.

Since ancient times, the matriarchal figure has provided unpaid care to men, women, children and the elderly, passing on their knowledge to subsequent generations. The health practices of primitive nomadic groups aimed at preserving man’s survival, a task then linked to the female sex [5]. During the Middle Ages, the role of care was centered on the role of healers, women considered wise among the feudal population, later called witches by the Church during the Inquisition and commonly linked to the status of sex workers and widows [5,6]. Considering the growing influence of the Church in the world, care was no longer provided by these groups and was in the hands of religious women in charity hospitals, in small, poorly organized groups [5,7].

In the 19th century, Florence Nightingale, from an English aristocratic family, launched the scientific and curricular bases of Modern Nursing, and revolutionized care in wartime contexts, highlighting the importance of cleaning, water, air purification, food and nutrition to achieve the cure of the sick [5]. Her religious upbringing shaped her practice and discourse in a dogmatic way, with a strong moral, patriarchal, obedience and discipline connotation, highlighting honesty, sobriety, religiosity and devotion as attributes of a reliable nurse, and favoring centrality in the biomedical model [8].

Nowadays, the male supremacy in science and in the world of work reinforces the profession’s stereotypes and collaborates for its social devaluation. For instance, the difficulty of women in accessing formal leadership positions, with a trajectory marked by challenges such as prejudice, team resistance, division of labor in the maternity ward, little investment in social capital [9,10]. An example of this difficulty is demonstrated in Brazil, where although the president of the Federal Council of Nursing is a woman in the current administration, this had not happened since 2009 [11]. In addition, the social, racial and technical division of labor reproduces inequalities of gender, race and class, following the intersectionality debate [12,13] of the current structure of power. Finally, the monetarily and social devaluation perpetuated by stereotypes of the profession. The last reinforces the subordination of nursing in relation to other health professions, especially medicine [7,14].

Foucault wrote about power relations and how they appear, or are forbidden, in discourses and how this same discourse reinforces these relations. In this way, it is power that determines whether the discourse will exist, will be silenced, or even if it will be taken as true. The rules of this dispute take place in the social arena and discourse analysis must understand it as plural, the result of a complex plot [15]. In this sense, it is possible to reflect on nursing education and the insertion of these professionals in a neoliberal labor market, as well as who these professionals are and what this represents.

In contemporary times, the movement to transform nursing curricula grows and gains more space, aiming to include philosophical and pedagogical aspects, but reductionist, fragmented and technicist teaching still prevails [16,17]. In addition to the content, pedagogical practices are still, for the most part, traditional and fragmented, without prioritizing the development of the student’s critical sense [18]. In this reality, themes such as gender, social power relations and revolutionary movements have little space for debate.

Critical reflection on this theme makes it possible to understand who these professionals are and how the power relationship to which they are submitted has developed. Intersectionality is a central concept in this discussion. 

The understanding of intersectionality arises from the Black women’s movement and demonstrates that oppressions are interconnected by race, class and gender, constituting a "matrix of domination" that operates at all levels of social relations, from the individual to the socio-structural [19]. Is a theoretical and methodological reference that allows thinking about multiple exclusions and seeks answers and strategies to overcome these paradigms.

Considering the foundations of nursing history and the importance of critical and reflective formation in the unequal context of gender, race and class relations that structure society, a study was developed to understand how the relationships of gender, feminism and empowerment are experienced by nursing students at a Brazilian public university, based on the analysis of their discourses.

## 2. Materials and Methods

### 2.1. Participants

The students were selected through convenience sampling and the sample was closed due to theoretical saturation. The participants were invited to participate through an announcement made at the Faculty of Nursing and the meetings took place between August to September 2018. The demographic data of the final sample included students of different ages, racial identifications, sexual orientations and course stages. This study received ethical approval from the Research Ethics Committee of the State University of Campinas under opinion number 94938618.0.0000.5404. 

### 2.2. Study Design

This is a qualitative, exploratory-explanatory study, with an approach aligned with Foucauldian discourse analysis. This type of study aims to understand the way in which the human being is interpreted, whether through history, relationships, representations, beliefs, perceptions and/or opinions. It is a level of reality that cannot or should not be quantified, which delves into the world of meanings [20,21]. 

According to Foucault, discourse should be understood in the plural and attention should be on what was said concretely, in a context and space of time. In this way, each discourse has its particularities, which must be considered, not in the search for the occult, but in understanding that they create a possible reality, mediated by exclusion and interdiction, control and selection, organization and distribution procedures. This methodological path highlights power relations and seeks to master the randomness of events [15].

Aiming at the methodological rigor of the study, the Consolidated Criteria for Reporting Qualitative Studies (COREQ) checklist was used. 

### 2.3. Procedure

Participants were recruited at a Public University, located in the state of São Paulo, Brazil. The students were selected after expressing interest based on an announcement made at the Faculty of Nursing with the consent of the local coordinator. The study objectives and data collection procedure were clarified and those who voluntarily agreed to participate signed the Informed Consent Form before the interviews. The interviews were conducted in a private room at the time and place chosen by the participant, and they were audio recorded to posterior transcription. All recordings and transcriptions were made with the consent of the participants and stored in Microsoft Office files (Word and Excel), with restricted access, ensuring the protection of data ownership.

At the beginning of the interview, the participants were informed of their right to abandon the research at any time without suffering any consequences. Each interview was conducted with students individually, without losing sight of their belonging to the social group and lasted a maximum of 68 minutes and a minimum of 22 minutes. 

Data collection was conducted through open interviews, mediated by the triggering question: “What does it mean to you to be a woman in nursing?”, to provide the identical stimulus for all participants and allow free association on the topic. From this question, the researcher guided the interview depending on each interviewee’s response [21], focusing on the participant’s experiences during their undergraduate period. The interview approached questions related to the gender debate on the curriculum, the students’ feelings when they experienced situations that perpetuated the profession’s stereotypes, whether they had examples of situations that addressed the theme, and if gender relations interfered in training.

### 2.4. Data Analysis and Feedback

The method of analysis chosen, consistent with the need to understand the perception of the population studied, was the Discourse of the Collective Subject (DCS). The DCS consists of the qualitative representation of collective thought, aggregating the analogous manifestations of different people in a single synthesis-discourse, written in the first person singular. It is based on the Theory of Social Representations and aims to rescue socially shared ideas, aiming to cover the various conceptions and ideas that appear in the discourse of a given social representation [22].

The DCS is based on three operators: the key expressions extracted from the interviewees’ discourse; the main ideas summarizing the content of a category of key expressions; and the discourse of the collective subject, which synthesizes the content of social representation [22].

The interviews were conducted by the first author of the paper (I.C.N), who was previously prepared by the coordinator of this study (D.S.S). Once the interviews were transcribed, to ensure consistency of procedures across all interviews, the two authors above analyzed the texts and defined the key expressions separately. The themes that emerged were then discussed and analyzed together to determine the final central ideas and the construction of the discourses.

Thus, the DCS represents the interviewees’ speeches. However, they are a unique synthesis for each theme identified in the key expressions and central ideas, that is, a collective social representation.

## 3. Results

Twelve nursing students participated, ten women and two men, aged between 17 and 28 years old. The characteristics of the group are detailed in Table 1.

From the analysis of the transcribed interviews, the most prevalent key expressions in the speeches were selected, allowing the central ideas to be highlighted for the construction of the Discourse of the Collective Subject, as described in Table 2. 

Part of this research findings has been previously published with a focus on the neglection of gender in nursing education [23]. Nursing curriculum remains centered on the traditional scientific model and strengthening stereotypes aimed at the feminization of the profession. In the current manuscript, we advance analysis in a feminist and intersectional perspective to understand how the relationships of gender, feminism and empowerment are experienced by nursing students, beyond discuss strategies toward empowerment in nursing education.

Following, for each of the central ideas, the DCS was built from the key expressions addressed by the participants.

### 3.1. Profession: Feminine and Stigmatized

“The fact that nursing is mostly female comes from a historical construction, from the time of Florence and even before, at the time of the nuns and prostitutes. […] I think the devaluation of the profession is a reflection of the devaluation of women in society, today and in the past. Society was built on sexist foundations and this reflects directly on our profession, which is feminine. This and the construction of the profession with traditional, religious precepts, collaborates with the stereotypes that we see until today, of ‘doctor’s assistant’, ‘prostitute’, ‘good and charitable’. For example, the stereotype that the nurse does everything for love, that she doesn’t care about her salary, that she can do a 98-h shift… it doesn’t care, because ‘she’s there to serve’, this religious, almost canonized nurse thing. […] either we are crazy, or we only serve to give injections, or we do everything for love, gift, art and vocation and never for study and science”.

### 3.2. Education: Distant from Gender Relations

“Every time we talk about gender in college, it was never focused on the profession, but on the patient we were going to take care of. Yet, this approach has always been superficial. […] It’s even funny, because it’s not even that people don’t talk about it… it seems that people simply deny its existence, because it seems that it’s not interesting for anyone to talk about it. The places where I found this debate were always external to the college, such as the Academic Center, Leagues (mainly Obstetrics), conversation circles, always outside the graduation […] among teachers, nobody is prepared to talk about it and it forms a vicious circle of denial. In addition, instead of deconstructing stereotypes, teachers end up reinforcing them by dictating dress rules or how students should behave, whether or not they are in the work environment. What does the way I dress have to do with my technical and relational skills as a nurse? It seems that the nurse can never get out of line, she always has to be perfect”.

### 3.3. Perspective: Empowerment of the Profession

“It frustrates me a lot that nursing, although it’s the second largest professional category in the country, simply cannot organize itself politically. Anyway, I think the scenery is positive. I think that the new generations of nurses, who have more contact with the feminist movement, mainly, will make the change. Because feminism brings this, right? Empowerment. I need to recognize myself as a worker, as a woman, as a nurse, in an area where there is already a stigma attached to you. There is no way to separate the empowerment of nursing from feminism, for all that we have said here. If feminism seeks equality and that women are not devalued for just being a woman, there is no way that a profession which is devalued precisely because of it walks away from feminism. […] Although I am optimistic, we must also be aware that the current political scenario does not contribute to this struggle, no. You can see that the time to fight for rights will come soon, it’s already on our doorstep”.

Considering an intersectionality bias, the collection of racial and sexual orientation data was important to analyze if there were any different perceptions among the students. Despite the chosen method being the Discourse of the Collective Subject, subsequent analysis showed that some highlights should be cited considering these differences.

During the interviews, men and women had very similar speeches about the difficulty of talking about gender in graduation and agreed that nursing training tends to perpetuate some stereotypes. As the initial question asked about being a woman in nursing, the men interviewed highlighted the feelings they heard from their classmates and showed that there was a constant stereotype that male nurses were homosexuals, as described in the first collective discourse. This same speech about the reproduction of stereotypes referring to sexual orientation was also evidenced in the speeches of students of different sexual orientations.

Considering white and non-white people, only one student declared herself to be Black, stated that she felt that Black women in nursing were even more invisible and that she felt that some prejudices were more evident when she was with white colleagues. Some interviewees, white and non-white, mentioned the invisibility of Black women in the history of nursing, considering that the main international references ignore the trajectory of nurses beyond the European continent, citing as an example the nurse Mary Jane Seacole, a contemporary of Florence Nightingale, who had a similar path but is less known.

## 4. Discussion

Nursing is directly associated with care that, historically, is considered a female practice. Although this characteristic has been described since antiquity, the sexual division of nursing practices was only instituted at the end of the 19th century, under the influence of Florence Nightingale, who structures modern nursing on moral and religious foundations that are perpetuated to the present day [24].

The sexual division coexists with the social division of work in nursing and medicine, since the dichotomies between art and science, vocation and study, intellectual work and manual work, were not formed based only on the dual relationship of gender, but also on issues referring to race, social class, religion, age and others that make up the complex social network [24]. In Brazil, data show that the profile of nursing reflects the formation of Brazilian society, marked by racial and social inequalities [25].

Foucault argues that power is given through relationships and that it is not only repressive, but rather holds a productive power aimed at the body, not to mutilate it, but to train it [26]. In this way, the relations of gender, race and class inequality that historically permeate nursing have a burden of responsibility for shaping the knowledge and practice of the profession, as evidenced by the students’ discourse and daily observed in social relationships that perpetuate gender stereotypes, such as: doctor’s assistant, prostitute, authoritarian, kind, submissive, angelic, devout, homosexuals, among others.

The impact of the absence of gender debate on the profession remains a powerful obstacle for nursing in a neoliberal society that places significant value on what can be counted and measured and on any kind of workforce that improves effectiveness and bottom line [27]. The dichotomy between medical practices (treatment) and nursing practices (care) clearly translates the male power over the female, being materialized in the devaluation of nursing in the social sphere and reflected in continuous working hours and lower wages [28]. This dual and socially “opposite” relationship evidences inequalities of gender, race and social class, because in a capitalist society whose organizing principle is the sexual division of labor, nursing is located as a “natural activity of women”. This scenario, in addition to reinforcing stereotypes and the marginalization of female nurses, also contributes to the perpetuation of this biomedical model aligned with the hegemonic neoliberal system that seeks to treat diseases instead of directing efforts to develop a healthy society.

According to research that analyzed the national nursing profile [25], 85.1% of professionals are women and 53% are Black. As for the technical and social division of work, 23% of professionals are nurses and 77% are not graduates. Among Black women, 57.4% have a high school level, while just over 37% are nurses. These data reinforce the importance of these professionals reflecting about intersectionality.

If liberalism is related to the unequal explorations of power and capital, one of its consequences is the focus on education as an object of care and attention for the formation of subjects adapted to this political reality [29]. The security device, defined by Foucault, is the technology of power used to produce and manage populations, based predominantly on neoliberalism [30].

Therefore, the relationship established between nursing and gender is shown to be dominant in the technical, political and social segregation of work, as well remembered by the students and reinforces the breadth of this device that reaches the social fabric, involving all aspects of life to serve this market logic.

This means that the social relations of power that define the history of the profession are the same capable of transforming it, in the sense of enabling empowerment through nursing education. For this purpose, a necessary exercise is that of denaturalization, which considers that things were constructed, therefore, they have not always existed this way. As proposed by Foucault, problematize rather than generalize [26].

For Nogueira et al. [23], teaching models and professor training needs to be improved focusing on new educational philosophies and pedagogical didactics, with an integral and constant movement of changing educational paradigms in order to stimulate critical reflections and strengthen new concepts for the students. A non-problematic training uses the formal educational institution for individuals with behavior that influences and enables the functionality of the social structure; it is a precarious subjectivity fruit of disciplinary power.

The teachers, in addition to not inserting the critical debate of gender in the disciplines, also end up perpetuating stereotypes with the students. Souza [24] states that the stereotypes of the profession are reproduced by teachers, who are also nurses, due to their professional experiences marked by unequal and stigmatizing gender relations, reinforcing the dominant paradigm.

The collective discourse of the students reinforces the need to include the discussion of gender in the nursing curriculum, as well as to present the theme in a transversal way throughout the course by revisiting the curricular plans and Pedagogical Political Projects of the institutions, in order to articulate the sociopolitical commitment to the real and collective interests of the majority population [31]. 

Foucault states that where there are power relations, there is a possibility of resistance. Under certain conditions and based on its own productive and inventive logic, it can be modified [26,32].

The approximation of the training of nursing professionals to the discussion of gender and feminism, regarding the historical construction of the profession [14,33] and the current challenges of overcoming inequalities, constitutes a global need towards the formation of empowered professionals, capable of sustaining the autonomy and social relevance of nursing [34].

Although Simone de Beauvoir, a feminist icon, only defined Feminism in 1980 as “an individual way of life and collective struggle”, the feminist movement has been present in society at least since the 19th and early 20th centuries [35]. The movement went through four waves. The first arose in Europe and the United States in the struggle for voting rights for women. The second started in the 1960s and ended in the 1990s, working for equal education and employment opportunities, maternity leave, birth control and abortion rights, as well as against domestic violence and culture of rape. The third wave problematizes the intersectionality of gender, race and social class and extends from the 1990s to the 2000s. The fourth wave started around 2012 and expands the range of feminist agendas, mainly due to the internet. The year 2015 was known worldwide as the year of the “feminist spring”, in which the movement gained a voice in the lives of women across the globe. In the context of fluid and instantaneous relationships in the cybernetic and globalized world, activists defend and disseminate thoughts and actions of the feminist struggle for social and professional equality, advocating for the end of sexual violence and the culture of rape. This has had an ascending impact on the dissemination of information and political achievements against misogyny [36,37].

The relationship between nursing and feminism is considered, for some authors, an “uneasy alliance”. As a profession, nursing was little affected by the first wave of feminism, but the agenda emerged from the second wave, when the struggle for access to education, professional training and freedom was more intense. While one sector of organized militancy urged youth interested in health care to shun nursing in favor of the higher and more lucrative status of medicine, other nurse leaders sought to encourage and support women in pursuit of careers, equal pay, fair social treatment, and rejecting attacks on nursing as “inferior women’s work” [38].

The insertion of the generation of students since birth in the culture of digital information reflects in its speech the connection with feminism driven by social networks of digital technology and materialized in experiences of political collectives of the university in spaces outside the undergraduate curriculum. The consequences of this insertion in the nursing profession are still little discussed in nursing practice, but they demonstrate the potential to revolutionize health care from a feminist perspective.

Created from the students’ perception of a mostly female profession, the students’ discourse reinforces that feminism is essential in the struggle for the empowerment of women nurses, where “empowerment” means “taking power”, autonomy, and freedom of choice, in different contexts of unequal power relations. In this way, the students of the new generation, who actively participate in the “fourth feminist wave” and discuss the agenda daily, have the potential to transform the trajectory of nursing, subverting the hegemonic paradigm of male supremacy.

Despite the feminist agenda and the discussion of gender being evidenced as essential for students, it is evident that the dialogue between gender, race and class in the discourse is still almost non-existent, evidencing the neglect of structural inequities that permeate society and the history of profession. This interlocution takes shape and deepens with the concept of intersectionality debated by important intellectuals of Black feminism, whether on the national scene through the voices of authors such as Sueli Carneiro, Lélia Gonzales and Djamila Ribeiro, or internationally with authors such as Patricia Hill Collins, Angela Davis, bell hooks and Kimberlé Crenshaw [23].

The topic deepens the discussion on how the discriminatory systems reproduced by patriarchy, racism and class oppression are structuring of unequal positions of gender, race and social class and how there are synergistic interactions of oppression of groups that accumulate these discriminatory markers, as proposed by intersectionality [39,40]. Although it is gaining more space in feminist debates today (despite having been introduced more emphatically in the third feminist wave) and, as perceived, still not present in student discourse, intersectionality is defended as an analytical tool capable of examining how the interdependence of power influences relationships and interactions in a society marked by plurality and inequalities, as well as the individual experiences of the subjects who are part of it [13].

Rescuing the trajectory of nursing from the perspective of intersectionality makes it possible to analyze, explain and understand the complexity of the relations of power and inequality that permeate the profession, seeking alternatives in professional training and practice that seek to break the discrimination of the profession in its multiple facets.

Priority and urgent strategic actions have been already proposed in a previous study [23], which focus on professor training, review of Pedagogical Political Project (PPP) and curriculum, implementation of emancipatory pedagogical practices and constant questioning of the obstacles to management and leadership faced by women. To this, we add that these actions should ideally converge to the creation of an emancipatory culture. In this new environment, students, professors, researchers, and staff would engage in open discussions to actively explore means to address inequalities and propose solutions together. Beyond embedding this cultural change in the institution’s DNA, it would also create an echo within the community, since individuals could transpose their social awareness to other environments.

In a look at the institutional context in which the nursing course is inserted, it is worth noting that the recent affirmative action policies—in progress at the aforementioned university since 2017 [41,42] ethnic-racial quotas and indigenous entrance exams—have produced a series of transformations and generated demands that mobilize and challenge professors and students around the themes of inclusion and diversity. In this sense, the current study produces relevant orientation markers for internal policies and also external to the nursing course, for recognizing disparities and confronting them at the micro and macro political level. The participation of the academic nursing community in broad forums at the university in question and beyond, which debate and deliberate on issues of gender, race and class, is fundamental for structural changes to develop and promote necessary ruptures in higher education institutions toward social equity.

Although the present study demonstrates progress in the gender debate in nursing education, it is important to note its limitations. The research was conducted at a public university, which is a space recognized for its political struggles. In addition, it provides a degree of flexible workload that enables articulating formal training with extracurricular activities that promote social discussions. In Brazil, nursing training in the face-to-face modality comprises 87.8% of courses offered in private institutions [43]. In addition, the advancement of the use of the virtual learning environment has been increasingly used, prioritizing the technical training of nurses to the detriment of critical and reflective training and serving the interests of the market.

## 5. Conclusions

The discourse of nursing students showed that professional training is still marked by conservative traditionalism, with the reproduction of gender stereotypes and devaluation of the profession. During graduation, the debate on the social context of gender is little addressed and often neglected, triggering the inadequacy of the curriculum and unpreparedness of the agents of professional training in nursing. For students, the empowerment of women in times of globalization and the rapid dissemination of information allow greater access to the guidelines of the feminist movement. The empowerment of nursing is only possible with the approximation of this movement, but there is still a gap in the discourse of students regarding the dialogue between gender, race and class.

The study reaffirms the importance of unveiling historically structured ideological mechanisms and their influences on current social dynamics as an essential strategy in the struggle for nursing that is recognized and socially valued. In this sense, the discussion and construction of gender awareness must be present in a transversal way in academic training if we want nursing to write its history in new ways.

## Figures and Tables

**Table 1 ijerph-20-00535-t001:** Characteristics of the participants.

Id	Gender	Age	Race	Sexual Orientation	Course Stage
1	Female	17	Black	Bisexual	3rd
2	Female	22	White	Heterosexual	2nd
3	Female	25	White	Homosexual	1st
4	Female	18	Brown	Bisexual	4th
5	Female	23	Black	Bisexual	5th
6	Female	27	Brown	Heterosexual	2nd
7	Female	25	Brown	Heterosexual	3rd
8	Female	20	Brown	Heterosexual	1st
9	Female	20	White	Bisexual	1st
10	Female	24	Brown	Heterosexual	5th
11	Male	22	White	Homosexual	4th
12	Male	21	Brown	Heterosexual	2nd

**Table 2 ijerph-20-00535-t002:** Central ideas from the key expressions.

Main Key Expression	Central Idea
“Stereotypes Pervading the Profession” “Majority Female Profession” “Florence Nightingale Influence” “Gender-Related Devaluation”	Profession: feminine and stigmatized The profession suffers from gender discrimination, mainly due to historical factors.
“Teacher’s unpreparedness and denial” “Importance of the theme transversality” “Perpetuation of Stereotypes in the Course” “Traditionalism permeating the course”	Education: distant from gender relations Traditional training and with teachers unprepared to address gender issues.
“Professional empowerment inseparable from feminism” “Unfavorable political situation” “Students as potential for change” “Lack of political involvement”	Perspectives: empowerment of the profession Optimism in the students’ potential for change as they demonstrate greater political engagement, despite the unfavorable political situation.

## Data Availability

Not applicable.

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
