# Peer review of "Gender and Empowerment by Nursing Students: Representations, Discourses and Perspectives"

_ijerph, 2022, doi:10.3390/ijerph20010535_

Round 1

Reviewer 1 Report

This manuscript aimed understand how the relationships of gender feminism and empowerment are experienced by nursing students during their formation at a Brazilian public university, based on the analysis of their discourses. This theme is relevant and urgent and meet the scope of the scientific journal. Although the study is of good quality, this reviewer considers that more information is needed to understand the methods and results.

A first aspect that draws attention is the existence of an article by the same authorship (1st and 6th authors), published in a Brazilian journal in 2021, with very similar objectives and methods, and the same results (https://doi.org/10.1590/0034-7167-2020-1001 - reference 24). The authors should cite the already published article more openly, indicating how and to what extent the manuscript under evaluation advances and deepens the debate on gender in nursing.

The title does not fully express the results presented. The authors report their intention to address gender and empowerment in nursing education. But, in addition to the fact that the interviewed participants are nursing students, training is not the focus, and little is addressed throughout the text.

The Introduction section is brief and contextualizes the problem that motivated the study. The announced objective is clear, feasible and consistent with the methods, but little aligned with the results presented. Like the title, the objective expresses a focus on the training process, but the authors are unable to deepen this aspect.

The Materials and Methods section sufficiently describes the methodological steps. It is only suggested that the authors inform if the interviews were restricted to only one guiding question or if other questions were asked to the participants.

Regarding the results section, the authors could indicate whether there were very different perceptions among the subjects, especially between male and female, white and non-white students, and different sexual orientations. Was there anything that stood out?

The Discussion section could be more in-depth. The authors could return to the idea of discourse and address Foucauldian constructs more vehemently, such as power and institutionalization, especially when addressing training, considering that the school, as a formal space of education, is a disciplinary institution that, to a large extent, seeks to shaping subjects, adapting them to society's expectations (at least those that dictate social, moral, and ethical norms, and hold power). Wouldn't educational institutions, then, be producers of oppression and segregation and, therefore, an important actor in the legitimation of gender, race/color, and class disparities?

It is also necessary to suggest alternative strategies to improve the identified scenario, providing bases for policymakers, and the limitations of the study.

Reviewer 2 Report

This is an interesting paper and an interesting area of study. However, there are a number of issues that need to be addressed.

Page 1 line 45 refers to the influence of gender on the scope of nursing practice. The authors don’t discuss how gender influences the actual scope of practice. Do they mean scope of practice? If so, this point needs more explanation.

The background contains a number of overgeneralizations. The paragraph beginning on line 50 speaks to the unpaid care women have given since ancient times. This point would be strengthened with a reference. In addition, what about matriarchal societies in ancient times?

This paragraph also states that before modern nursing, 2 main groups provided care for the sick, and describes Catholic and Anglican religious (religious members? Nuns?) as “socially discredited.” Members of Catholic and Anglican churches have not been around since “ancient times.” In addition, there has been nursing care in many countries and societies around the world where there were no Catholics or Anglicans, or where other people besides “widows” and “prostitutes” provided care. Suggest substituting the term sex workers for prostitutes.

The paragraph beginning on line 65 refers to the “disqualification” of the nursing profession. This point is vague and unclear, and lacks evidence. How is the profession “disqualified”?

This same paragraph reports that most leadership positions in nursing are held by men. This is not true and there is no evidence for this point. Perhaps this is the case in Brazil? It is not the case around the world, and if the authors are only referring to Brazil, this point needs to be clear.

The end of the paragraph notes the profession is in “latent devaluation” and subordinate to other health professions. What current evidence is there to substantiate the claim that nursing is subordinate to the other health professions? In addition, if it actually is subordinate, this would not be a “latent” devaluation, but quite obvious.

The paragraph beginning on line 80 states that the education system for nursing continues to be fragmented and technicist. This is not the case in many countries. If this is the case in Brazil, this is a different point, and this should be clarified. In addition, many nursing programs in North America, UK and Europe discuss gender and power relations as part of the nursing curriculum.

Consider capitalizing the B in Black when referring to race/ethnicity.

With respect to the methods, why did the researchers collect race and sexual orientation data? How was this included in the analysis? In addition, why is the category race seemingly limited to three colours? And is this data needed?

Only one interview question is given and it is described as “triggering.” Why is it described as triggering? Were participants told it was supposed to be triggering? What other types of questions were asked?

The data regarding the participants’ characteristics would fit better in the results section.

The results section largely consists of 3 separate long quotations. There is also a chart presented. Could more information be provided about the findings and the analysis?

On page 7, lines 268-269, it states that we are still in the 3rd wave of feminism. Later, on line 279, it states we are in the 4th wave. Agree with the authors that we are in the 4th wave, and this could be consistent.

In addition, the first wave is often considered the suffrage movement – not the French Revolution as described in lines 264-266. However, if this is considered the first wave in other countries, this point could be highlighted.

Round 2

Reviewer 1 Report

I congratulate the authors for the manuscript proposal and for the improvements undertaken. I am pleased with the opportunity given to me to read this work in advance and evaluate it.

Reviewer 2 Report

Thank you for making these edits, and for providing the explanations.